# The Association of Erythropoietin and Age-Related Macular Degeneration in Hemodialysis Patients: A Nationwide Population-Based Cohort Study

**DOI:** 10.3390/ijms23179634

**Published:** 2022-08-25

**Authors:** Evelyn-Jou-Chen Huang, Fung-Chang Sung, Peir-Haur Hung, Chih-Hsin Muo, Meei-Maan Wu, Chih-Ching Yeh

**Affiliations:** 1Department of Ophthalmology, Taipei Medical University Hospital, Taipei 110, Taiwan; 2Department of Ophthalmology, School of Medicine, College of Medicine, Taipei Medical University, Taipei 100, Taiwan; 3Department of Health Services Administration, China Medical University, Taichung 406, Taiwan; 4Management Office for Health Data, China Medical University Hospital, Taichung 406, Taiwan; 5Department of Food Nutrition and Health Biotechnology, Asia University, Taichung 413, Taiwan; 6Department of Internal Medicine, Ditmanson Medical Foundation Chia-Yi Christian Hospital, Chiayi 600, Taiwan; 7Department of Applied Life Science and Health, Chia-Nan University of Pharmacy and Science, Tainan 717, Taiwan; 8Graduate Institute of Clinical Medical Science, College of Medicine, China Medical University, Taichung 406, Taiwan; 9Department of Public Health, School of Medicine, College of Medicine, Taipei Medical University, Taipei 110, Taiwan; 10School of Public Health, College of Public Health, Taipei Medical University, Taipei 110, Taiwan; 11Master Program in Applied Epidemiology, College of Public Health, Taipei Medical University, Taipei 110, Taiwan; 12Department of Public Health, College of Public Health, China Medical University, Taichung 406, Taiwan; 13Cancer Center, Wan Fang Hospital, Taipei Medical University, Taipei 116, Taiwan

**Keywords:** erythropoietin, age-related macular degeneration, hemodialysis

## Abstract

This population-based retrospective cohort study investigated the effectiveness of erythropoietin (EPO) treatment in reducing the risk of age-related macular degeneration (AMD) in hemodialysis patients, using the National Health Insurance Research Data of Taiwan. From the database, we identified 147,318 end-stage renal disease (ESRD) patients on hemodialysis who had been diagnosed in 2000–2014 to establish the propensity-score-matched EPO user cohort and non-EPO user cohort with equal sample size of 15,992. By the end of 2016, the cumulative incidence of AMD in EPO users was about 3.29% lower than that in non-EPO users (Kaplan–Meier survival *p* < 0.0001). The risk of AMD was 43% lower in EPO users than in non-EPO users, with an adjusted hazard ratio (aHR) of 0.57 (95% confidence interval (CI) = 0.51–0.64) estimated in the multivariate Cox model. A significant negative dose–response relationship was identified between the EPO dosage and the risk of AMD (*p* < 0.0001). Another beneficial effect of EPO treatment was a reduced risk of both exudative AMD (aHR = 0.48, 95% CI = 0.40–0.61) and non-exudative AMD (aHR = 0.61, 95% CI = 0.53–0.69), also in similar dose–response relationships (*p* < 0.0001). Our findings suggest that EPO treatment for hemodialysis patients could reduce AMD risk in a dose–response relationship.

## 1. Introduction

Age-related macular degeneration (AMD) is an important disorder leading to irreversible visual impairment and severe central vision loss in the elderly. Many risk factors have been associated with AMD, including smoking, hypertension, obesity, and sunlight exposure [1,2,3]. Clinically, AMD is a three-stage progressive disorder. In the early stage, medium-sized drusen and retinal pigmentary changes are present. In the intermediate stage, the patient may suffer from blurry or wavy vision. In the late stage, the patient may suffer from vision failure due to neovascularization and non-neovascularization [4]. Dietary antioxidant supplementation may slow progression from early AMD to late AMD. The major treatment for neovascular (also known as wet or exudative) AMD is intravitreal injections of anti-vascular endothelial growth factor (anti-VEGF) agents [5]. For non-neovascular (known as atrophic, dry, or non-exudative) AMD, no proven therapies are currently available; several agents are being investigated in clinical trials [6].

Erythropoietin (EPO) is a hypoxia-induced hormone that is primarily produced in adult kidneys through inhibition of the apoptosis of red blood cell precursors, benefiting the increase in red cell mass. Studies have found that both EPO and VEGF were increased in the vitreous fluid of patients with ischemic retinal diseases such as proliferative diabetic retinopathy (PDR) and central retinal vein occlusion (CRVO) [7,8]. Anti-VEGF agents are currently the main effective drugs used to reduce neovascularization of various retinal diseases. EPO may play a protective role in angiogenesis such as inhibiting hypoxia-inducible factor-1alpha (HIF-1) expression and preventing angiogenesis [9]. Furthermore, experimental studies have reported that EPO could protect the inner blood–retinal barrier [10], the outer blood–retinal barrier [11] and neuronal cells [12] in early diabetic retinopathy. Therefore, EPO is proposed to play a beneficial role in neuroprotective effects through the mechanisms of anti-ischemic, anti-inflammatory, and antioxidant effects in retinal vascular diseases [13,14]. EPO can be used as a new therapeutic agent for retinal degeneration due to pro-angiogenic signaling molecules at the molecular level under oxidative stress [9,15]. Using intravitreal EPO to treat atrophic AMD [16], non-arteritic anterior ischemic optic neuropathy [17], acute vascular occlusion of the retina posterior pole [18], and diabetic macular edema [19] is a new strategy, but with limited data. Hence, the treatment effectiveness of intravitreal injection of EPO deserves further investigation.

Synthetic EPO has been successfully used to correct anemia for patients with ESRD since the first use in 1987 [20]. Except for the correction of anemia, EPO treatment in chronic renal failure patients improves depression, fatigue, muscle power, exercise ability, and neurocognitive function. On the other hand, a meta-analysis of nine trials with 5143 patients reported that EPO treatment may increase the risk of death, myocardial infarction, stroke, and venous thromboembolism when higher hemoglobin concentrations (hemoglobin levels over 120–160 g/L) were targeted [21]. However, evidence pertaining to the effects of EPO use in ESRD patients on the risk of AMD development remains limited. In this study, we used insurance claims data from Taiwan to conduct a population-based study to investigate the effectiveness of EPO for the treatment of AMD in ESRD patients. We also examined whether EPO treatment has an effect on AMD risk in a dose–response pattern.

## 2. Results

After propensity score matching, 15,992 EPO users and 15,992 non-EPO users were identified for inclusion in the study cohorts. Distributions of all baseline variables were similar in both cohorts (Table 1). In both cohorts, over half of the ESRD patients were older than 70 years and lived in urban areas. The top three comorbidities were hypertension, diabetes, and anemia.

The cumulative incidence of AMD in the EPO cohort was 3.29% lower than that in the non-EPO cohort (Figure 1). During the follow-up periods, the AMD incidence in the EPO cohort (10.52 per 1000 person-years, or 668 cases) was almost half of that in the comparisons (20.38 per 1000 person-years, or 647 cases) by the follow-up time (Table 2). Compared to non-EPO users, the HR of AMD was 0.57 (95% CI = 0.51–0.64) for EPO users in both crude and adjusted Cox models (Table 2). The AMD risk decreased as the EPO dosage increased (*p* for trend < 0.0001). The aHRs of AMD decreased from 0.69 (95% CI = 0.57–0.82) at EPO dosage level Q1 to 0.49 (95% CI = 0.41–0.58) at level Q4, compared to non-EPO users.

The cumulative incidences of exudative AMD and non-exudative AMD in EPO users were approximately 0.18% and 3.20% lower than those in non-EPO users, respectively (Figure 1). Incidence of exudative AMD and non-exudative AMD in EPO users were significantly lower than in non-EPO users (1.84 vs. 5.10 and 8.68 vs. 15.3 per 1000 person-years), respectively, the corresponding aHRs were 0.48 (95% CI = 0.40–0.61) and 0.61 (95% CI = 0.53–0.69), respectively (Table 3). The risks of exudative AMD and non-exudative AMD were also significantly reduced as the EPO dosage increased (*p* for trend < 0.0001).

Table 4 presents the analysis of hemodialysis patients treated with EPO and non-EPO users by follow-up duration. The incidence of AMD in EPO users was about half of that in non-EPO users within the first five years. The corresponding aHR in less than 1 year, 1.0–1.9 years, 2.0–2.9 years, 3.0–3.9 years, and 4.0–4.9 years were 0.54 (95% CI = 0.44–0.66), 0.47 (95% CI = 0.36–0.62), 0.48 (95% CI = 0.35–0.65), 0.49 (95% CI = 0.36–0.67), and 0.58 (95% CI = 0.37–0.90), respectively. However, the decreased risk of EPO on AMD was no longer observed after 5 years (aHR = 1.02, 95% CI = 0.63–1.71).

## 3. Discussion

Prior studies investigating the development of AMD associated with chronic kidney disease are limited. In 2016, a study from the NHIRD of Taiwan reported that patients with ESRD are at a higher risk of developing AMD than people without kidney disease, and the risk is higher in PD patients than in HD patients [22]. Results of a Korean study also reported that chronic renal disease is associated with not only a higher risk of peripheral drusen, but also developing AMD in 3008 participants aged 50–87 years [23]. Based on these studies, it could be concluded that there is a possible risk of AMD developing in patients with chronic renal disease or ESRD. Unfortunately, these studies have not explored the effectiveness of EPO therapy associated with the development of AMD.

To the best of our knowledge, this is the first study exploring EPO therapy’s effectiveness in reducing the risk of AMD for ESRD patients, using real-world data from Taiwan. Our findings revealed that EPO treatment was effective in reducing the risk of developing AMD for 43% of EPO users.

EPO is a glycoprotein hormone which facilitates red blood cell production in humans. Studies have shown that EPO and EPO receptors (EPOR) in the retina play physiological roles relating to the treatment of ocular disorders. EPO may play a role of beneficial neuroprotective effects including anti-ischemic, anti-inflammatory, and antioxidant effects in retinal vascular diseases [13,14]. Studies on diabetic retinopathy found that EPO is released in response to tissue damage and appears to have protective functions [24].

AMD is a multifactorial disease. One of the possible mechanisms leading to AMD is neurodegenerative condition, such as lipofuscin accumulation in retinal pigment epithelium (RPE) cells due to impairment of the autophagy system responsible for RPE cells in AMD patients. Because of neuro and tissue-protective mechanisms, EPO may inhibit apoptosis, thereby leading to a beneficial effect in the treatment of retinal disease [25]. EPOR has been found in ocular RPE, photoreceptors, horizontal cells, ganglion cells, amacrine cells, and bipolar cells, expressing neuroprotective or other effects in these cells [25,26,27]. A case series study has found that EPO therapy could improve visual function for patients with traumatic optic neuropathy [28].

Another common underlying mechanism is associated with angiogenesis. Exudative AMD is characterized by choroidal neovascularization, and it is well known that VEGF plays an important role in the growth of abnormal blood vessels. However, studies have shown that VEGF does not complete angiogenesis, and EPO is a significant independent factor for the development of angiogenesis [29,30]. The hypoxia-inducible factor-1alpha (HIF-1α) can activate VEGF for angiogenesis [31]. With the regulation of HIF-1α, the binding of EPO/EPOR leads to new blood vessel formation [25].

AMD and PDR have the same character of up-regulated angiogenic reactions through VEGF due to ischemia status. The intravitreal injection of anti-VEGF drugs is the current gold standard treatment for both AMD and PDR. It has been noted that intravitreal Ranibizumab (anti-VEGF) injection in PDR patients could reduce the expression of EPO [32]. However, EPO concentration increased in the vitreous fluid from ischemic retinal diseases and could play a role in the pathogenesis of PDR [33].

In our study, EPO users among hemodialysis patients had lower risks of developing AMD compared to non-EPO users. However, the role of EPO in the etiology and pathogenesis of AMD remains under debate. A recent study clarified that EPO therapy may play a dual switch neuroprotective mechanism of antiapoptotic and pro-angiogenic determinant, protecting the retina under up-regulated VEGF [15]. Other studies provided different findings implying that the EPO may play a protective effect under the hypoxic status [25]. It is well known that hypoxia-upregulated EPO can provide protection to reduce apoptosis of erythroid progenitors. Laboratory studies showed that EPO can treat retinal disease for adult mice through the inhibition of apoptosis [25]. Hemodialysis patients have insufficient EPO production because of poor kidney function in general, and EPO treatments may protect these patients from developing AMD based on the aforementioned experimental results.

This study is strengthened by using two large population-based data sets to establish the propensity-matched cohorts for a long follow-up period to explore EPO treatment effectiveness on reducing the AMD risk in ESRD patients. Nevertheless, some insufficiencies in our study should be addressed. First, diseases were identified based on ICD-9 codes in the claims data. The accuracy of diagnoses may raise concern. However, the National Health Insurance Bureau of Taiwan maintains a system to conduct regular cross-checking of diagnoses. The claims data have been used in previous studies and have been proven valid, including those for AMD [34,35,36,37,38]. Second, information on education, lifestyle, visual acuity, family history of AMD, and environmental factors, which might potentially contribute to or protect individuals from AMD, were not available in the databases. However, we were able to use multivariate Cox models to calculate adjusted values controlling for age, gender, and all comorbidities to minimize the potential confounding effects. Third, the level of ferritin and transferrin saturation (TSAT) used to adjust the EPO treatment in dialysis patients were also unavailable in our data. However, the insurance requires care providers to maintain serum ferritin at levels > 100 ng/mL and/or TSAT at levels of >20% for patients during EPO therapy. Fourth, only ESRD patients with hemodialysis were included in this study, the treatment effectiveness of EPO for peritoneal dialysis patients deserves study. Fifth, the availability of continuous EPO therapy for patients with ESRD should be considered. Patients undergoing hemodialysis in Taiwan are entitled to use EPO treatment as long as they have anemia. Therefore, the integrity of EPO treatment should be appropriate. Finally, this study lacks the exploration of endogenous EPO levels for ESRD patients. Studies analyzing serum EPO levels in patients with CKD have shown that ESRD patients with severe endogenous EPO deficiency, but there is no impact on Hb levels [39,40]. Moreover, an investigator-initiated prospective study has shown that 300 μg of darbepoetin alfa could increase serum EPO levels by 130–270-fold in the first 24 h, compared with no darbepoetin alfa, and remained significantly elevated after 3 days [41]. Since there is no correlation between endogenous EPO and Hb levels in ESRD patients, and the level of exogenous EPO is much higher than that of endogenous EPO, it is appropriate to study the effect of exogenous EPO on AMD.

## 4. Materials and Methods

### 4.1. Data Sources

For a retrospective cohort study, we used claims data in the National Health Insurance Research Database (NHIRD) available at a Health and Welfare Data Science Center (HWDSC) in Taiwan. Taiwan has set up a single-payer mandatory enrollment National Health Insurance program since 1 March 1995, and up to 99% of the population has enrolled in this program. NHIRD contained the registry for beneficiaries, inpatient claims, outpatient claims, and the registry of catastrophic illness. Data were linked to each other by de-coded personal identification number for protecting privacy of patient. International Classification of Diseases, Ninth Edition, Clinical Modification (ICD-9-CM) was used to identify the disease before 2016, and ICD-10-CM was used since 2016 in the databases. WHO Anatomical Therapeutic Chemical Classification System (ATC code) was used to classify the use of medicine. This study proposal has been approved by the Research Ethics Committee of China Medical University and Hospital in Taiwan (CRREC-107-021).

### 4.2. Study Subjects

We identified 147,318 ESRD newly diagnosed patients with hemodialysis in 2000–2014 from NHIRD at HWGSC. Those with peritoneal dialysis (PD) history, kidney transplant history, malignancy history, myopia history, AMD history, transfer to PD, age younger than 18 years, deceased at ESRD diagnosis date, or without information of living area were excluded. From the remaining 103,877 ESRD patients, we found 87,757 patients had received EPO treatment and 16,120 patients had not received the treatment.

We calculated the propensity score for each person by logistic regression with an adjustment for sex, age, urbanization level, ESRD-diagnosed date, and comorbidities. An EPO-users cohort and a propensity-score-matched comparison cohort of non-EPO users were established with equal sizes using greedy algorithms. Figure 2 presents the detailed selection process.

### 4.3. Demographics, Comorbidity, EPO Exposure, and Outcome

The information on demographics available in the claims data included age, gender, and urbanization level (urban, suburban, and rural). According to Liu’s report, urbanization was classified into 7 levels [42]. Levels 1 and 2 were designated as “urban” areas, levels 3 and 4 the “suburban” areas, and levels 5–7 the “rural” areas. Comorbidities that we considered as potential covariates that might be associated with the development of AMD included cardiovascular disorders such as coronary heart disease, hypertension, atrial fibrillation, and diabetes, etc. All comorbidities were defined before the ESRD-diagnosed date. ESRD patients who received EPO treatment during the follow-up period were defined as users. The dosage of EPO per week (defined daily dose, DDD) was calculated: the sum of EPO use in the study duration divided by the sum of follow-up week. We divided the EPO dosage into four levels by quartile (Q1 level: 1–948 DDDs, Q2 level: 949–2974 DDDs, Q3 level: 2975–10,630 DDDs, and Q4 level: >10,630 DDDs). Follow-up person-years were calculated for both cohorts starting from the ESRD-diagnosed date to the diagnosis of AMD, death, or the end of 2016. AMD was grouped into two subtypes: exudative type (ICD-9-CM codes 362.42, 362.43, 362.52, and 362.53) and non-exudative type (ICD-9-CM codes 362.50 and 362.51).

### 4.4. Statistical Analysis

All variables were presented with numbers and percentages, compared between EPO user and non-EPO user cohorts. The standardized difference was estimated to show the balance of variables between two cohorts, the standardized difference over 0.1 was considered a significant difference. We calculated the incidence of AMD using the number of individuals diagnosed with AMD divided by the follow-up duration (person-year) for both cohorts. Univariate and multivariate Cox proportional hazards regression models with time-dependent covariate were used to calculate the EPO cohort to the comparison cohort crude hazard ratios (cHRs) and adjusted hazard ratios (aHRs) of AMD, respectively. The aHRs were calculated after controlling for sex, age, and all comorbidities in the multivariate Cox models. The association between EPO treatment and AMD risk was assessed not only by the EPO dosage, but also by the AMD subtype. To observe the dose–response relationship between EPO dosage and risk of AMD, the linear trend test was used to estimate the potential trends. We used the Version 9.4 SAS software (SAS Institute Inc., Cary, NC, USA) for data analyses. Unless otherwise specified, a *p* value of <0.05 was considered statistically significant.

## 5. Conclusions

Hemodialysis patients could have benefited from EPO treatment with lowered risk of subsequent development of AMD, including the subtypes of both exudative and non-exudative AMD. Significant dose–response effectiveness was observed in reducing the risk of AMD as the EPO dosage increased. Further investigations are needed to clarify the underlying patho-mechanism between the development of AMD and EPO usage in ESRD patients.

## Figures and Tables

**Figure 1 ijms-23-09634-f001:**
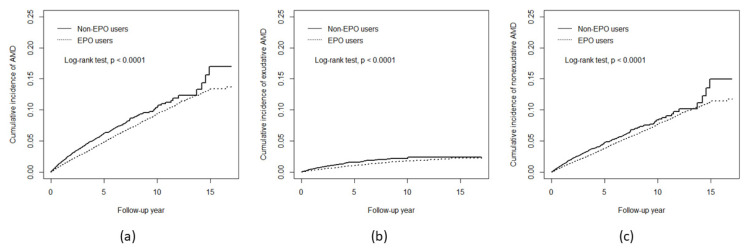
Cumulative incidence for (**a**) AMD, (**b**) exudative AMD, and (**c**) non-exudative AMD in two EPO cohorts.

**Figure 2 ijms-23-09634-f002:**
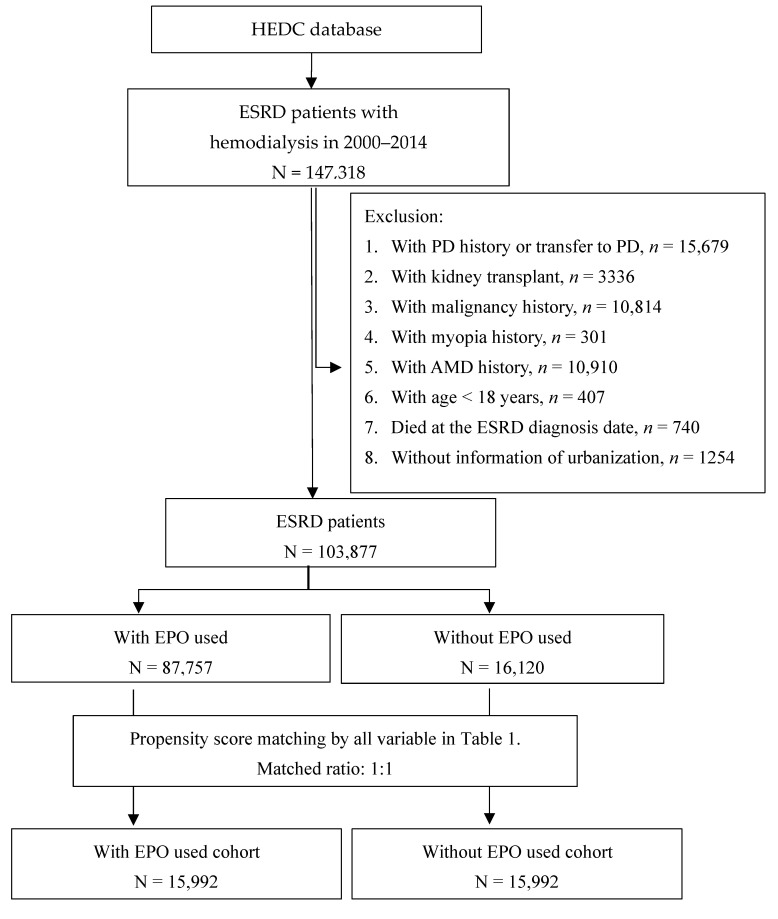
Flow chart of study cohort selection.

**Table 1 ijms-23-09634-t001:** Demographic characteristics and comorbidities compared between cohorts with and without erythropoietin treatment in hemodialysis patients in Taiwan.

	EPO Users		Non-EPO Users		
	N = 15,992		N = 15,992		Standardized
Variable	*n*	%	*n*	%	Difference
Age (year)					
18–40	295	1.84	336	2.10	0.018
41–50	928	5.80	995	6.22	0.018
51–60	2293	14.3	2115	13.2	0.032
61–70	3917	24.5	3742	23.4	0.026
≥71	8559	53.5	8804	55.1	0.031
Gender					
Male	8119	50.8	8069	50.5	0.006
Female	7873	49.2	7923	49.5	0.006
Urbanization					
Urban	8459	52.9	8465	52.9	0.001
Suburban	5211	32.6	5206	32.6	0.001
Rural	2322	14.5	2321	14.5	0.000
Comorbidity					
Coronary heart disease	8457	52.9	8419	52.7	0.005
Hypertension	14,711	92.0	14,719	92.0	0.002
Diabetes	10,839	67.8	10,830	67.7	0.001
Atrial fibrillation	1477	9.24	1435	8.97	0.009
Heart failure	6796	42.5	6863	42.9	0.008
Hyperlipidemia	6956	43.5	7007	43.8	0.006
Anemia	9186	57.4	9230	57.7	0.006
Cataract	5939	37.1	6031	37.7	0.012
Diabetic retinopathy	3575	22.4	3587	22.4	0.002
Stroke	6918	43.3	6900	43.2	0.002
Viral hepatitis	1545	9.66	1580	9.88	0.007
DDDs/week, median (IQR)	2974.7	(9681.8)			
Follow-up years, mean (SD)	3.97	(3.61)	1.02	(2.10)	1.000

DDDs, defined daily doses; EPO, erythropoietin; IQR, interquartile range; SD, standard deviation.

**Table 2 ijms-23-09634-t002:** Incidence and hazard ratios of age-related macular degeneration in hemodialysis patients by quartile dosage of erythropoietin.

EPO (DDDs/Week)	Event	PY	Rate *	cHR (95% CI)	*p*	aHR (95% CI) ^†^	*p*
Non-EPO Users	647	31,748	20.38	Ref.		Ref.	
EPO users	668	63,514	10.52	0.57 (0.51–0.64)	<0.0001	0.57 (0.51–0.64)	<0.0001
Q1	157	12,812	12.25	0.67 (0.56–0.80)	<0.0001	0.69 (0.57–0.82)	<0.0001
Q2	180	14,358	12.54	0.68 (0.57–0.80)	<0.0001	0.71 (0.60–0.84)	<0.0001
Q3	161	18,116	8.89	0.49 (0.41–0.58)	<0.0001	0.49 (0.41–0.58)	<0.0001
Q4	170	18,228	9.33	0.51 (0.43–0.60)	<0.0001	0.49 (0.41–0.58)	<0.0001
*p* for trend				<0.0001		<0.0001	

CI, confidence interval; DDDs, defined daily doses; EPO, erythropoietin; Q, quartile; HR, hazard ratio; PY, person-years. * Incidence rate, per 1000 person-years; ^†^ Adjusted for age, gender, urbanization level, and all comorbidities.

**Table 3 ijms-23-09634-t003:** Incidence and hazard ratios of subtype age-related macular degeneration in hemodialysis patients by quartile dosage of erythropoietin.

EPO (DDDs/Week)	Exudative AMD			*p*	Nonexudative AMD			*p*
	Event	Rate *	aHR (95% CI) ^†^		Event	Rate *	aHR (95% CI) ^†^	
Non-EPO users	162	5.10	Ref.		485	15.3	Ref.	
EPO users	117	1.84	0.48 (0.40–0.61)	<0.0001	551	8.68	0.61 (0.53–0.69)	<0.0001
Q1	27	2.11	0.57 (0.38–0.87)	0.008	130	10.15	0.72 (0.59–0.88)	0.001
Q2	31	2.16	0.59 (0.40–0.87)	0.008	149	10.4	0.74 (0.62–0.90)	0.002
Q3	32	1.77	0.45 (0.31–0.66)	<0.0001	129	7.12	0.50 (0.41–0.61)	<0.0001
Q4	27	1.48	0.36 (0.24–0.54)	<0.0001	143	7.84	0.53 (0.44–0.64)	<0.0001
*p* for trend			<0.0001				<0.0001	

CI, confidence interval; DDDs, defined daily doses; EPO, erythropoietin; Q, quartile; HR, hazard ratio; PY, person-years. * Incidence rate, per 1000 person-years. ^†^ Adjusted for age, gender, urbanization level, and all comorbidities.

**Table 4 ijms-23-09634-t004:** Incidence and hazard ratios of age-related macular degeneration in hemodialysis patients treated with erythropoietin by follow-up duration.

Follow-Up Year	Non-EPO Users		Rate *	EPO Users		Rate *	aHR (95% CI) ^†^
	Event	PY		Event	PY		
<1.0	300	10,872	27.6	154	11,611	13.3	0.54 (0.44–0.66)
1.0–1.9	125	6159	20.3	99	10,395	9.52	0.47 (0.36–0.62)
2.0–2.9	81	4340	18.7	80	8863	9.03	0.48 (0.35–0.65)
3.0–3.9	56	3021	18.5	70	7281	9.61	0.49 (0.34–0.70)
4.0–4.9	31	2107	14.7	52	5900	8.81	0.58 (0.37–0.90)
5.0+	54	5248	10.3	213	19,463	10.9	1.02 (0.63–1.71)

CI, confidence interval; EPO, erythropoietin; HR, hazard ratio; PY, person-years. * Incidence rate, per 1000 person-years. ^†^ Adjusted for age, gender, urbanization level, and all comorbidities.

## Data Availability

The Taiwan Nation Health Insurance Database was used with permission and perform data analyses at the Taiwan Health and Welfare Data Science Center. Restrictions apply to the availability of these data, which were used under license for this study. Information about using the data base is detailed in https://dep.mohw.gov.tw/DOS/np-2497-113.html (accessed on 10 April 2022).

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
