# Peer review of "The Association of Erythropoietin and Age-Related Macular Degeneration in Hemodialysis Patients: A Nationwide Population-Based Cohort Study"

_ijms, 2022, doi:10.3390/ijms23179634_

Round 1

Reviewer 1 Report

Authors in the manuscript titled “The association of erythropoietin and age-related macular degeneration in hemodialysis patients: a nationwide population-based cohort study” used claims data of Taiwan and investigated the association of erythropoietin (EPO) with age-related macular degeneration (AMD) in patients receiving hemodialysis.

The manuscript is interesting to show the lower incidence of AMD in EPO users compared to non-EPO users among patients with hemodialysis. However, there are some points that need more clarification.

Major critics:

- The authors mentioned that EPO and VEGF were both increased in ischemic retinal diseases such as PDR and CRVO. VEGF produced in ischemic states lead neovascularization, so that anti-VEGF agents became main stream of treatment to reduce neovascularization of various retinal diseases. In similar context, EPO may not always play a beneficial role. Accordingly, I think that the sentence that suggests a beneficial role of EPO (lines 61-63) is not properly placed, as the former sentence mentioned that EPO is increased in ischemic conditions (i.e. suggesting a harmful effect such VEGF). That paragraph needs to be revised.

Minor critics due to typo errors

- “EPO and EPO receptors (EPOR) in the retinal” in line 147: should be ‘in the retina’.

- “hypoxia upregulated EOP” in line 180: should be ‘EPO’.

Reviewer 2 Report

In this manuscript, the authors concluded that EPO administration may be effective in preventing, to some extent, the development of AMD in patients undergoing hemodialysis for chronic renal failure, based on the observation that the incidence and risk of AMD, especially atrophic AMD, were significantly lower in patients receiving EPO and that there was a negative correlation between EPO dosage and the incidence of AMD. Although the results are relatively clear and the conclusions seem justified at first glance, this reviewer believes that the following points warrant further discussion. 

1.     Since patients with chronic renal failure receive EPO exogenously due to a decrease in endogenous EPO, the effects of EPO in the patient's body should be discussed based on the sum of both endogenous and exogenous effects. This paper does not examine endogenous EPO, and a discussion of this issue is warranted. Measurements of actual patients' blood EPO levels would be more desirable for this study.

2.     In addition, patients on hemodialysis often require oral phosphorus adsorbents to control serum phosphorus levels. Some adsorbents contain iron, which increases hematocrit, and the exogenous EPO dosage should be increased or decreased accordingly. Therefore, the dose of exogenous EPO is not always constant, but may vary depending on the patient's blood test data, and it may not be realistic to use a single data set for a single patient as in this study. 

Round 2

Reviewer 2 Report

The points raised in the previous peer review were well considered and the quality of the paper seems to have improved sufficiently.